# Composition Characteristics of an Urban Forest Soil Seed Bank and Its Influence on Vegetation Restoration: A Case Study in Dadu Terrace, Central Taiwan

Chia-Yen Lin [1], Min-Chun Liao [2] , Wei Wang [2] and Hsy-Yu Tzeng [1,*]

1 Department of Forestry, National Chung Hsing University, Taichung 40227, Taiwan; benny82626@gmail.com
2 Taiwan Forestry Research Institute, Taipei 10066, Taiwan; seedfolk@tfri.gov.tw (M.-C.L.);
  f3022002_5@hotmail.com (W.W.)
* Correspondence: erecta@nchu.edu.tw; Tel.: +886-422-840-345 (ext. 142)

**Abstract:** The contributions of urban forests and green spaces to sustainable development have been confirmed. Meanwhile, cities worldwide have shown that investments in urban forestry can greatly contribute to citizens' quality of life. This study was conducted in urban forests in the Dadu Terrace of Taichung City, central Taiwan, which were frequently disturbed by fires and had grassland severely invaded by *Panicum maximum* after the forest degraded. We sampled 46 plots in Dadu Terrace to understand the relationship between the soil seed bank and vegetation as well as to evaluate the feasibility of applying soil seed bank transfers for ecological restoration in Dadu Terrace. The grassland was dominated by *Panicum maximum*. Forest vegetation was distinguished by cluster analysis into five types, i.e., *Ficus microcarpa* type, *Acacia confusa* type, *Litsea glutinosa* type, *Cinnamomum camphora* type, and *Trema orientalis* type. In the aboveground survey, we recorded 141 vascular plants, including 129 seed plants and 12 ferns. There were 40 identified species of naturalized plants. A total of 29,914 seedlings were recorded in the soil seed bank, with an average seed density of 9634 seeds/m$^2$ and a total of 91 species. There were 40 species of naturalized plants, accounting for 90.9% of the total seed reserves. This showed that Dadu Terrace was severely affected by the invasion of naturalized species. The species number and seed reserves of woody plants of the *Panicum maximum* type were significantly lower than those of forest vegetation. The composition of the soil seed bank was dominated by naturalized plants, indicating that the high frequency of fire reduced the proportion of native species and woody plants in the soil seed bank. *Acacia confusa* type was the main forest type in Dadu Terrace. Although several woody species and seed reserves were in its soil seed bank, the naturalized proportions were even higher. *Trema orientalis* type was the secondary forest type in Dadu Terrace; it had the smallest forest area. However, it was the only vegetation type with a greater tree seed abundance than herbs and the lowest proportion of naturalized seed abundance. *Trema orientalis* type vegetation has a relatively high soil transfer value for ecological restoration but lacks diversity. Our results revealed that the characteristics of the soil seed bank of Dadu Terrace make it challenging to restore the grassland to the forest by natural succession. Therefore, we suggest that artificial restoration is necessary for Dadu Terrace.

**Keywords:** urban forest; soil seed bank; fire; naturalized plant; invasive plant; restoration

## 1. Introduction

Seed availability is a critical key to recovery [1,2], which is determined by seed production conditions, seed rain, and soil seed bank [3]. The species composition and seed reserve of soil seed banks can represent a specific restoration capacity [4]. They can also be used to describe vegetation succession mechanisms and trends in plant communities and predict the recovery of pioneer populations after disturbance [5–7]. Therefore, the soil seed bank is considered a crucial component of potential vegetation restoration, and soil seed bank composition can be applied to ecological restoration [8–10], thereby providing

an essential reference for reforestation and forest ecosystem management [5,9,11–13]. More and more studies have explored the role of seed banks in invasive species succession and vegetation restoration [14–17].

Western Taiwan has a flat terrain, and it has lost much of its natural forest in the low-elevation parts of the mountains owing to population growth and development of agriculture and industry. The Dakeng area and Dadu Terrace in Taichung City contain the remaining forests and play a critical role in urban forests in central-western Taiwan. The former is richer in terms of forest and species diversity [18,19], whereas the latter has planted forests with *Acacia confusa* as the main silvicultural species [5] (Figure 1a).

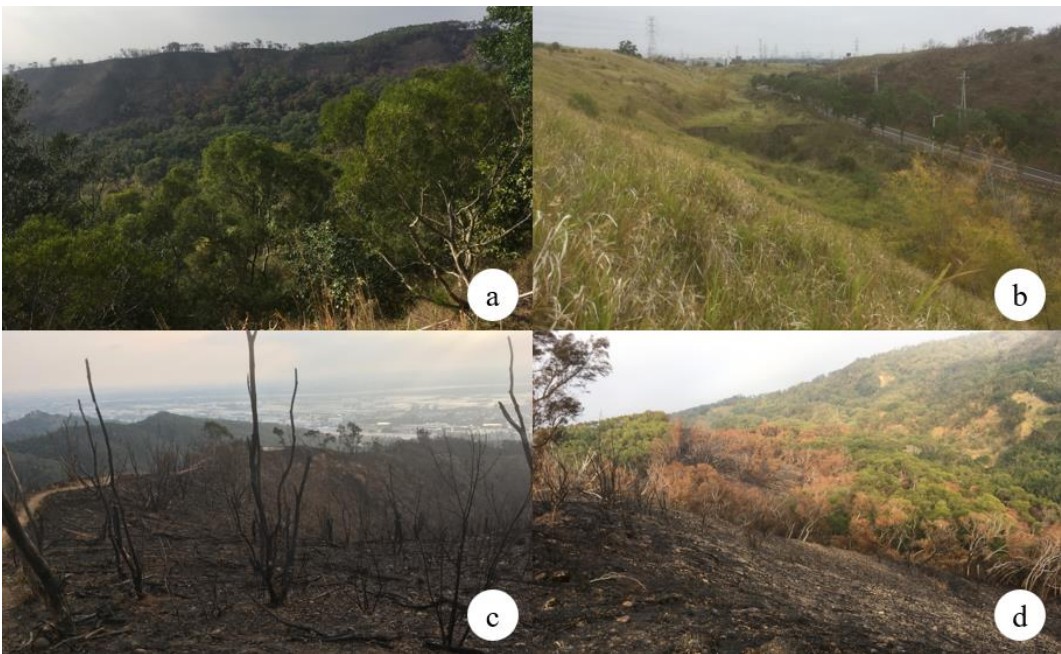

**Figure 1.** (**a**) *Acacia* forest; (**b**) *Panicum maximum* grassland; (**c**) burnt trees; (**d**) fire disturbance in Dadu Terrace of Taiwan.

A perennial native to Africa, *Panicum maximum* is widely cultivated in tropical and subtropical regions and is often used as pasture [20]. It is now one of the world's most invasive plants [21,22]. *P. maximum* was introduced to Taiwan in 1908 and was planted as a provision grass in Dadu Terrace during Japanese rule [23]. Because *P. maximum* has a higher photosynthetic rate than the native *Miscanthus sinensis*, it grows faster in the same environment [24]. Additionally, *P. maximum* grew rapidly after the fire (Figure 1b) [5,20,24] and has thus become one of the top 20 invasive plants in Taiwan [25], resulting in a savanna-like landscape in Dadu Terrace after the frequent fires (Figure 1c,d) [26]. When *P. maximum* invades and forms grassland, it disrupts the forest's structural composition and changes the composition of the soil seed bank [5].

In this study, we want to understand the composition characteristics of urban forest soil seed banks and their influence on vegetation restoration in Dadu Terrace. We analyzed the types of aboveground vegetation, seed reserves, and the relationship between aboveground vegetation and soil seed banks. Then, we compared with the ratio of naturalized plants, the number and seed abundance of tree species in soil seed bank characteristics among different plantation types to estimate the potential plantation and natural successional restoration trends in Dadu Terrace. We also assessed the feasibility of applying soil seed banks for ecological restoration as a reference for future ecological restoration of this urban forest.

## 2. Materials and Methods

### 2.1. Study Site

Dadu Terrace in Taiwan is located on the west side of Taichung Basin, bordered by Dajia River in the north and Dadu River in the south, with a length of approximately 20 km from north to south, a width of approximately 7 km from east to west (Figure 2a), and a maximum elevation of 310 m. Its geology is part of the Toukoshan formation, and the soil is mainly red clay with poor water retention and contains a lot of gravel [27,28]. The climate of Dadu Terrace is characterized by distinct wet and dry seasons. The climate diagram (Figure 2b) shows that the dry period lasts from October to January of the upcoming year and the per-humid period lasts from March to September. The forest area has gradually reduced in recent years owing to land development and fire disturbance [29]. The composition of the existing vegetation can be divided into forest and grassland based on the vegetation physiognomy, and the latter is dominated by the invasive plant *P. maximum*. One of the dominant species that form the forest is *A. confusa*, one of the essential afforestation species in Taiwan, and it has the largest area of the broadleaf plantation at low elevation [30]. In addition, there are many cemeteries in the area. After entering dry autumn and winter seasons, the aboveground *Panicum maximum* accumulates flammable fuels [31], which are often accidentally lit by human activities. The frequent fire disturbance has caused a reduction in the forest area of Dadu Terrace [29,32] (Figure 1c,d), resulting in a retrogressive succession of vegetation. As the deforested area expands and the surrounding forest vegetation becomes remote and fragmented, forest recovery becomes less resilient, resulting in slow or even stagnant succession.

### 2.2. Setting of Sampling Plot and Vegetation Survey

The study site was divided into three parts, north, central, and south regions, by the geography of Dadu Terrace. For each region, we selected several plant communities by vegetation composition and physiognomy characteristics, and 17 areas (A–R) were set up. Three plots were set up for each area, except for O_(1 plot), Q_(1 plot), and R_area (two plots) due to the small size of the plant community. We set 46 sampling plots, and the distance between plots was about 10–30 m within the area. Because the composition and structure of the Dadu Terrace forest were relatively simple and the canopy height of the forest is mostly below 10 m, the sampling plot size was set at 10×10 m. Each sampling plot was subdivided into four 5 × 5 m subsampling plots in which the aboveground vegetation was surveyed. We surveyed the frequency, covered area, and basal area of plant occurrence in the sampling plot and recorded all herbaceous plants, vines, and woody plants (with a diameter at breast height (DBH) < 1 cm) as an understory to count their coverage area. The forest sampling plot was also surveyed for the overstory of trees with a DBH of >1 cm; its DBH was recorded, and basal area was calculated. These characteristics were converted to relative frequency, relative coverage, or relative dominance, which were summed to the importance value (IV). The IV values of understory and overstory were calculated separately, i.e., the IV value of understory is relative frequency + relative coverage, and the IV value of overstory is relative frequency + relative dominance.

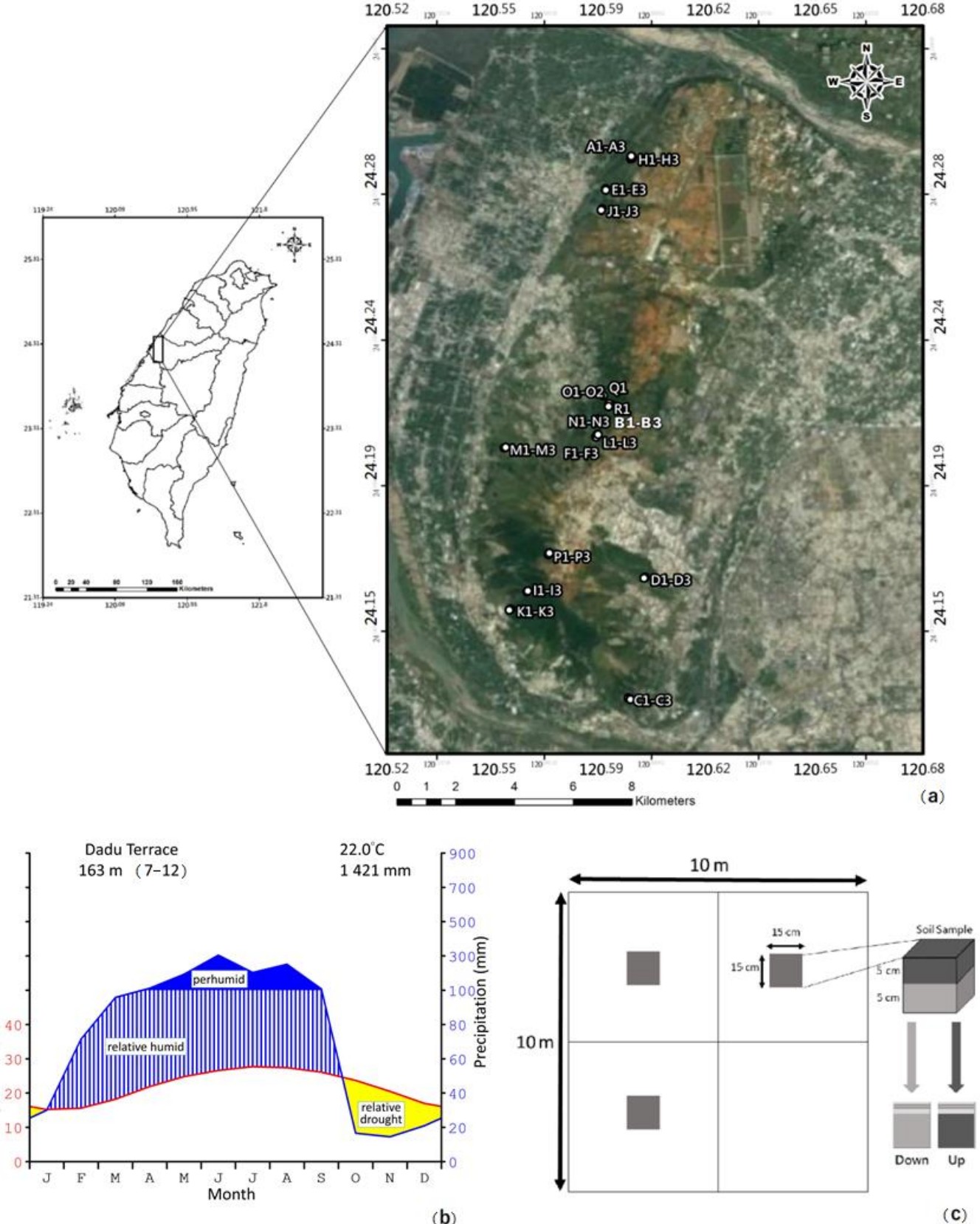

**Figure 2.** The information of this study: (**a**) Location of sampling plot; (**b**) climate diagram; (**c**) sampling plot design.

### 2.3. Sampling and Germination Test of the Soil Seed Bank

Soil seed bank samples were collected from the centers of the subsampling plots in the first, second, and third quadrants of each planting area randomly, then three 15 cm × 15 cm soil samples were taken from each area using a metal frame tool. The area was divided into two soil layers, the upper layer of 0–5 cm (including the layer of dead leaves) and the lower layer of 5–10 cm, which were marked and brought back to the laboratory in sealed zip-lock bags (Figure 2c). The soil samples were placed in black plastic bags immediately after collection and stored at room temperature without light to prevent early germination of seeds in the soil samples by light and temperature stimulation. Soil samples were collected in 276 bags from 20 to 26 March 2017, with a total soil sampling plot of 31,050 cm$^2$ and a soil sample volume of 310,500 cm$^3$.

In this study, the seedling emergence method was used to study the soil seed bank [3]. The experimental site was located at the Beigou Tree Nursery, Department of Forestry, National Chung-Hsing University, without a shade net in a greenhouse that could block most foreign seeds from entering. The collected soil samples were placed separately and evenly in a 50 cm × 30 cm × 6.5 cm plastic germination container lined with white nonwoven fabric. Furthermore, 1–2 cm perlite was placed under the nonwoven fabric to ensure that the seeds and soil would not be lost through drainage holes. Water was sprinkled regularly at three times (8:00, 12:00, and 16:00) daily for 3 minutes each time. The germination experiment was initiated on 27 March 2017 and ended on 17 August 2017, with 21 weeks of germination, and weekly observations were carried out to identify the germinating seedlings and record their numbers. The identified seedlings were removed so as not to block the germination of other seeds. The plant species that could not be identified were moved to other large pots, planted until they could be identified, and were then removed. Three control plastic germination containers were set up during the experiment, nonwoven and seed-free imported Akadama soil were laid to confirm whether foreign seeds were introduced into the soil samples.

### 2.4. Measurement of Environmental Factors

Environmental factors affect the growth and survival of plants and the causes of vegetation composition. Different soil conditions affect the growth of seedlings [33]. Environmental factors affect invasive plants' geographic distribution, and colonization is also an important issue [34]. The environmental factors measured in this study were elevation (Ele), slope (Slo), whole light sky (WLS), aspect (Asp), and moisture gradient (MG). The sampling plot was located by a global positioning system (GPS), and the elevation of the sample center was measured. The slope was measured directly using a compass to determine the elevation of the sampling plot. The whole light sky was measured using a compass to determine the elevation of 12 azimuths around the center of the sampling plot; the whole light sky refers to the size of the airspace in which the sampling plot can receive solar radiation, which is a comprehensive estimate of aspect, slope, terrain shading, and solar radiation energy [35,36]. Aspect refers to the direction the slope of the sampling plot faces, which is transformed to the corresponding value of moisture gradient [37].

In this study, three soil collection sites were randomly selected in the sampling plot, within which the upper layer of dead leaves was cleared first. After mixing the topsoil at a depth of approximately 10 cm, samples were placed at room temperature and air-dried. Samples were then sieved using a two mm sieve and measured for soil pH (Soil_pH) [38], total nitrogen (Soil_N) [39], total organic carbon (Soil_C) [40], available phosphorus [41], and cation exchange capacity (CEC) [42].

### 2.5. Data Analysis and Statistical Methods

The scientific names of plants were based on *Flora of Taiwan II* [43]. Rare plants were based on *The Red List of Vascular Plants of Taiwan, 2017* [44]. The naturalized plants are referred to in local documents in Taiwan [9,45,46]. The growth forms of seed plants were divided into four types, including trees, shrubs, vines, and herbs.

Principal coordinate analysis (PCoA) was conducted to understand the correlations between the aboveground vegetation and soil seed bank. The overstory IV values of the forest samples were via cluster analysis using the Sørensen similarity index and single linkage method. In addition, a gradient analysis of the understory IV of forest and grass was conducted to understand the correlation between the composition of ground cover species and environmental factors using detrended correspondence analysis (DCA) and canonical correlation analysis (CCA). In this study, the gradient of the axial length of the DCA results was more than two standard deviations, and if the axial length of DCA was greater than four, CCA was performed using environmental factors [47]. The statistical software PC-ORD 6 [48] was used for cluster analysis and ordination analysis.

The species and number of seedlings germinated in each soil sample were recorded, and the soil seed bank was compiled to showcase each species and its corresponding number. Because ecological data of soil seed banks are not normally distributed [49], the species composition and seed reserves of soil seed banks among plant communities were analyzed using the nonparametric Kruskal–Wallis test. A post hoc assessment was performed to compare the differences in the composition of native and naturalized species in different plant communities. This part was computed by SPSS 22.0 [50].

We used the Sørensen similarity index [51] to investigate the similarity in species composition of the soil seed bank among different plant communities.

$$SI = 2c/(Va + Sa)$$

where SI is the Sørensen similarity index, and Va and Sa are the number of species for aboveground vegetation and soil seed bank in the same plot_a, respectively. Moreover, c is the number of species occurring in aboveground vegetation and soil seed banks. The Sørensen similarity index was calculated by R 4.1.2 version and Simba package.

## 3. Results

### 3.1. Composition of Aboveground Plant Community

A total of 141 species of vascular plants, including 129 species of spermatophyte and 12 species of pteridophyte, were recorded. Among them, 100 native species and 41 naturalized species were included; the naturalized plants were all spermatophytes. Spermatophytes included 35 tree species, 27 shrub species, 25 vine species, and 42 herb species according to the type of growth. The top three families with the most species were Asteraceae (19 species), Euphorbiaceae (14 species), and Rubiaceae (8 species). One species—*Zanthoxylum avicennae*—was listed in *The Red List of Vascular Plants of Taiwan, 2017* as vulnerable (VU), and two species—*Acronychia pedunculata* and *Lindera glauca*—were data deficient (DD).

A total of 37 forest plant communities were sampled, cluster analysis was conducted based on the IV of tree species of the sampling plot, and a dendrogram was drawn (Figure 3). According to the vegetation physiognomy, the plant community was divided into forest and grassland. The plots of forest plant communities were classified into five types with information maintenance of 40% as the threshold value. Furthermore, the most dominant species in the basal area of the overstory were used as the names of the vegetation types (Figure 4), including *Ficus microcarpa* type (six sample plots), *Acacia confusa* type (two-sample plots), *Litsea glutinosa* type (six sample plots), *Cinnamomum camphora* type (four sample plots), and *Trema orientalis* type (one sample plot).

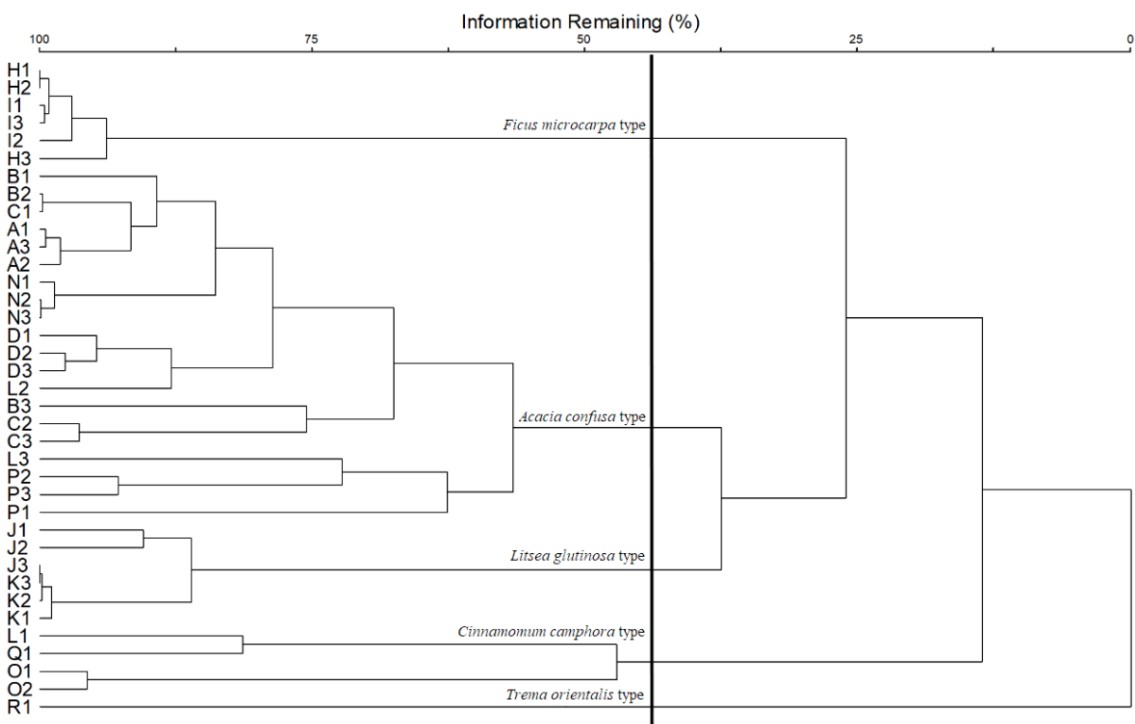

**Figure 3.** Cluster analysis of aboveground vegetation in Dadu Terrace, Taiwan.

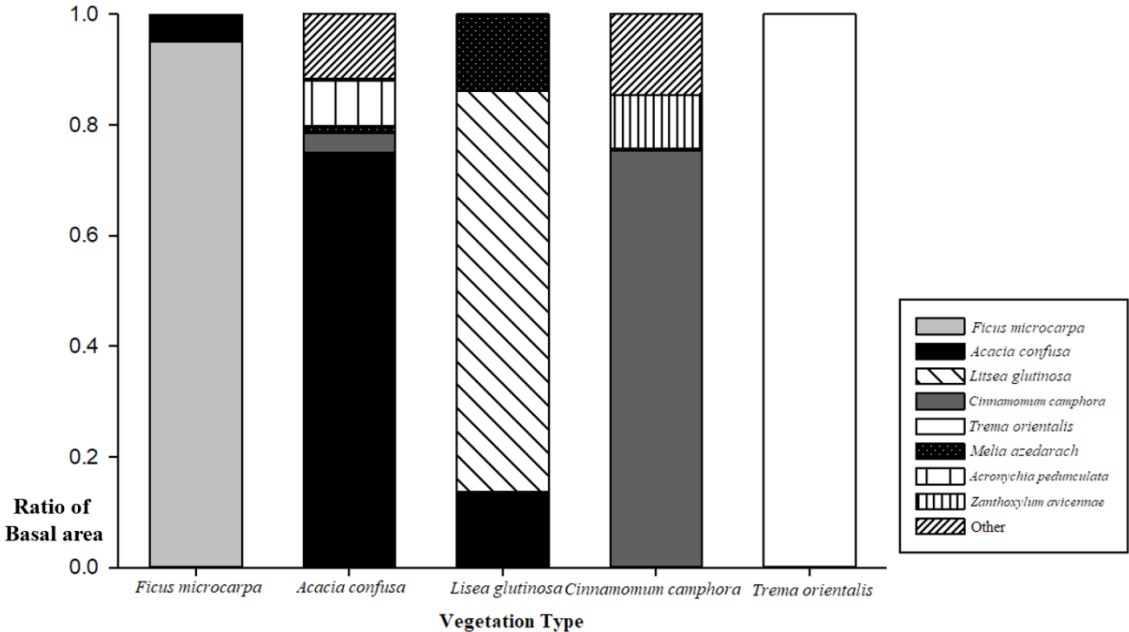

**Figure 4.** The basal area ratio with different vegetation types in Dadu Terrace, Taiwan.

The herb community was named after *P. maximum*, the most dominating plant in the sampling plot. In addition to *P. maximum*, there were some shrubs such as *Lantana camara* and associated annual–biennial plants such as *Bidens pilosa* var. *radiata* and *Praxelis clematidea*. Among the forest vegetation, all were plantations except the *Trema orientalis* type, a secondary forest. The sampling plots were dominated by *A. pedunculata*, *L. glauca*, and *Z. avicennae*, and other species were classified as *Acacia confusa* type owing to the sporadic presence of *A. confusa*. The number of species differed significantly among the aboveground vegetation types ($p < 0.05$). The post hoc results of the pairwise vegetation types showed that the number of species in the *Panicum maximum* type was significantly less than those in the

*Cinnamomum camphora* type and *Acacia confusa* type ($p < 0.05$) (Table 1). The proportions of species within the naturalized plants differed significantly from vegetation types ($p < 0.05$). *Panicum maximum* type was significantly higher than *Acacia confusa* type and *Cinnamomum camphora* type ($p < 0.05$).

**Table 1.** Summary of aboveground vegetation and soil seed bank conditions among different vegetation types in Dadu Terrace, Taiwan.

| Vegetation Type | Grassland | | Forest | | | |
|---|---|---|---|---|---|---|
| | *Panicum maximum* Type ($n = 9$) | *Acacia confusa* Type ($n = 20$) | *Ficus microcarpa* Type ($n = 6$) | *Litsea glutinosa* Type ($n = 6$) | *Cinnamomum camphora* Type ($n = 4$) | *Trema orientalis* Type ($n = 1$) |
| Species number of aboveground vegetation | 11.4 ± 3.8 [b] | 23.6 ± 11.2 [a] | 14.5 ± 10.0 [ab] | 17.2 ± 7.7 [ab] | 35.0 ± 6.7 [a] | 16.0 |
| Percentage of naturalized species of aboveground vegetation (%) | 56.7 ± 11.6 [a] | 26.3 ± 16.5 [b] | 30 ± 10.7 [ab] | 49.4 ± 18.7 [ab] | 23.1 ± 2.2 [ab] | 56.3 |
| Percentage of naturalized species covered of understory (%) | 96.8 ± 5 [a] | 51.5 ± 41.9 [b] | 45.6 ± 22.6 [ab] | 88.2 ± 12.1 [ab] | 10.3 ± 6.1 [b] | 97.1 |
| Average munber of species in soil seed bank | 13.8 ± 4.4 [a] | 15.1 ± 5.2 [a] | 14.8 ± 1.7 [a] | 15.7 ± 1.5 [a] | 13.8 ± 3.2 [a] | 10.0 |
| Average seed reserve in soil seed bank (seeds/m$^2$) | 771.0 ± 494.5 [a] | 523.5 ± 482.6 [a] | 618.3 ± 460.9 [a] | 748.8 ± 398.3 [a] | 928.5 ± 324.4 [a] | 589.0 |
| Average number of tree species in soil seed bank | 0.4 ± 0.5 [b] | 2.6 ± 1.4 [a] | 2.3 ± 0.8 [a] | 2.5 ± 0.5 [a] | 2.5 ± 1.5 [ab] | 1.0 |
| Average tree seed storage in soil seed bank (seed/m$^2$) | 0.7 ± 1.0 [b] | 19.2 ± 18.9 [a] | 6.3 ± 3.9 [ab] | 10.2 ± 10.4 [ab] | 7.0 ± 4.1 [ab] | 404.0 |
| Percentage of naturalized species in soil seed bank (%) | 73.0 ± 5.5 [a] | 56.4 ± 12.2 [b] | 60.3 ± 8.1 [ab] | 65.5 ± 7.7 [ab] | 52.4 ± 11.6 [ab] | 90.0 |
| Percentage of soil seed reserves of naturalized species (%) | 97.3 ± 2.3 [a] | 78.0 ± 21.9 [a] | 95.7 ± 2.6 [a] | 95.6 ± 4.2 [a] | 87.7 ± 11.8 [a] | 31.4 |
| The average Sørensen similarity index between aboveground vegetation and the soil seed bank | 0.28 ± 0.11 [a] | 0.20 ± 0.12 [a] | 0.16 ± 0.05 [a] | 0.50 ± 0.12 [a] | 0.19 ± 0.08 [a] | 0.31 |
| Range of Sørensen similarity index between aboveground vegetation and soil seed bank | 0.15–0.45 | 0.00–0.50 | 0.09–0.23 | 0.36–0.64 | 0.09–0.26 | 0.31 |

Kruskal–Wallis test: significance ($p < 0.05$). Significant differences are distinguished using superscript ab.

Comparing the coverage of understory plants, a significant difference was found among vegetation types ($p < 0.05$), and the ground coverage of naturalized plants of the *Panicum maximum* type was significantly higher than that of *Acacia confusa* type and *Cinnamomum camphora* type ($p < 0.05$) (Table 1).

The DCA results of the 46 samples of aboveground vegetation understory in Dadu Terrace showed that the total variation was 6.40; the eigenvalues were 0.678, 0.421, and 0.307 in the first three axes, where the explanation rates of variation were 10.6, 6.6, and 4.8% in the first three axes, and the lengths of the first three axes were 4.03, 3.171, and 2.86, respectively (Figure 5). The sampling plot of *Acacia confusa* type straddled the first and third quadrants of the DCA ordination diagram, the sampling plot dominantly containing *A. pedunculata* was distributed on the rightmost side of axis 1, and the sampling plots of *Acacia confusa* type and *Panicum maximum* type that dominantly comprised *P. maximum* were distributed in the third quadrant of the ordination diagram, whereas the sampling plots of *A. confusa* type in the middle of Dadu Terrace (B1–B3, L1–L3) were distributed between the aforementioned two. *Litsea glutinosa* type had a distinctive ground cover

composition owing to the dominance of *L. glutinosa* seedlings and was clearly distinguished from the other plant samples dominated by *P. maximum* in axis 2.

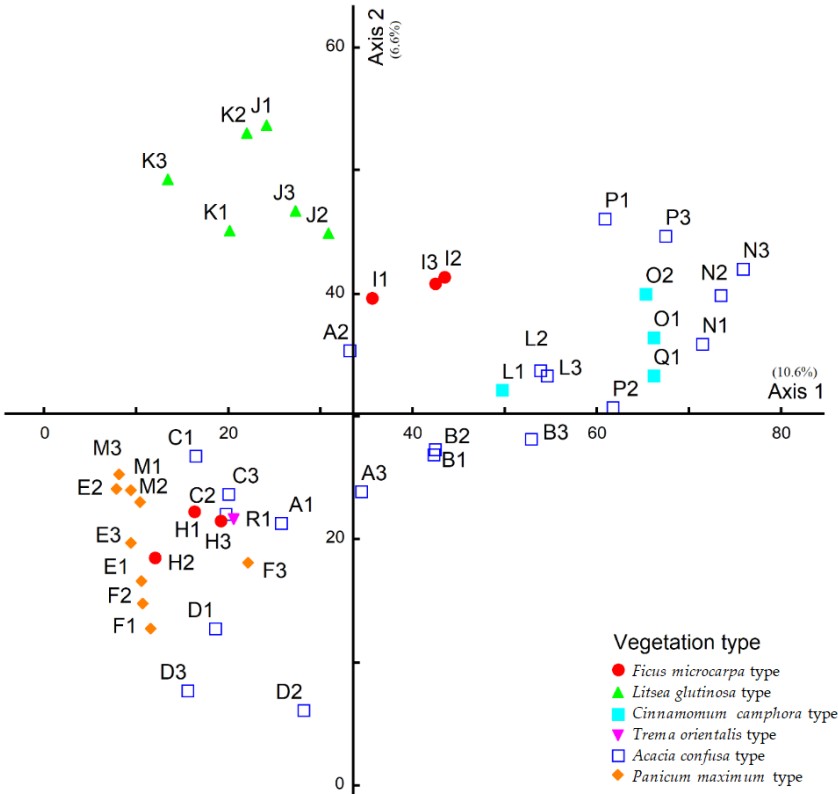

**Figure 5.** Detrended correspondence analysis (DCA) of understory vegetation in Dadu Terrace, Taiwan.

The CCA analysis results showed that the characteristic values of the first three axes were 0.528, 0.399, and 0.276, and the explanation rates of variation of the first three axes were 8.2, 6.2, and 4.3%, respectively. Among the environmental factors, moisture gradient, slope, soil total nitrogen, and soil_pH were the most significant ones (Figure 6). The sampling plot of *P. maximum* grassland was distributed on the leftmost side of axis 1 of the CCA ordination diagram, which showed that it had a higher soil_pH (4.78–6.64) and a lower soil total nitrogen (0.097–0.203%), whereas forest had a lower soil pH (pH 3.80–5.93) and a higher soil total nitrogen (0.122–0.542%). In addition, *Acacia confusa* type had a higher CEC (7.70–19.30 cmol/kg), which was mainly distributed in the right side of axis 1, whereas *Cinnamomum camphora* type had a higher moisture gradient (12.22–16.00), and the sampling plots were mainly distributed in the first quadrant.

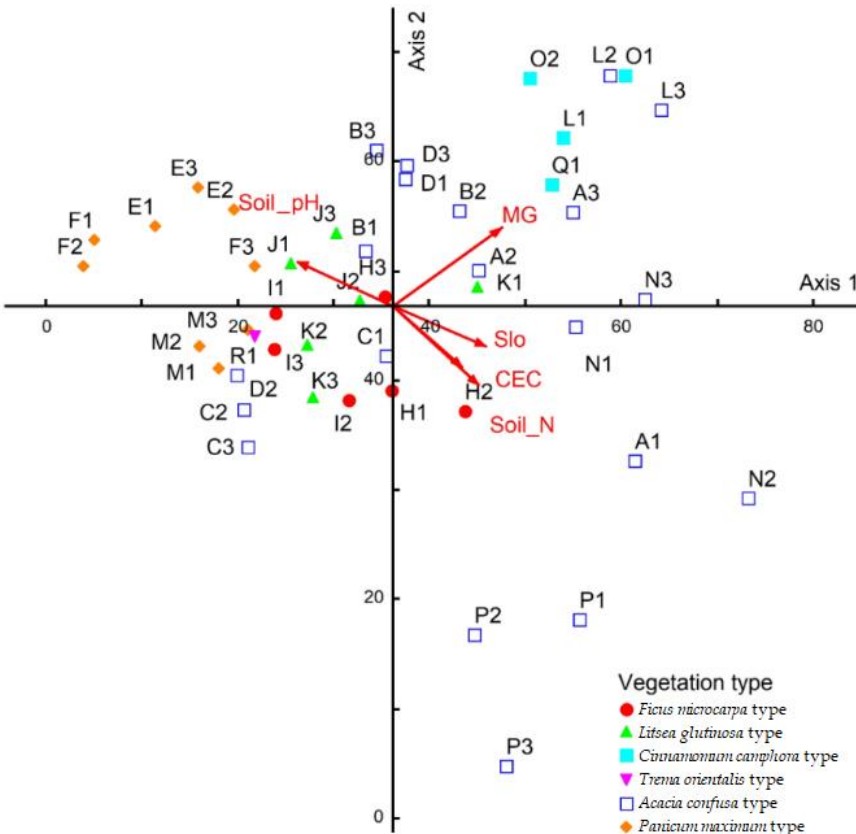

**Figure 6.** Canonical correlation analysis (CCA) of understory vegetation in Dadu Terrace, Taiwan. CEC: cation exchange capacity of soil; MG: moisture gradient; Slo: slope; Soil_N: soil total nitrogen; Soil_pH: soil pH.

### 3.2. Composition of the Soil Seed Bank

This study recorded 29,914 seedlings of soil seed banks, with an average seed density of 9634 seeds/m$^2$, containing 36 families, 79 genera, and 91 species. Within this set, the top three families with the highest number of species were Asteraceae (19 species), Poaceae (8 species), Cyperaceae (6 species), and Euphorbiaceae (6 species); the top three families with the most abundant seed reserves were Rubiaceae (5763 seeds/m$^2$), Asteraceae (2362 seeds/m$^2$), and Solanaceae (1007 seeds/m$^2$). The number of species and seed reserves of plant growth types were most dominated by herbs (51 species, 9170 seed/m$^2$), followed by shrubs (16 species, 49 seed/m$^2$), trees (14 species, 296 seed/m$^2$), and vines (10 species, 119 seed/m$^2$). The top 10 species in the soil seed bank accounted for 90.61% of the total seed reserves in the following order: *Spermacoce latifolia*, *Solanum americanum*, *Pr. clematidea*, *Pa. maximum*, *Kyllinga brevifolia*, *T. orientalis*, *Conyza sumatrensis*, *Oplismenus compositus*, *Mikania micrantha*, and *Soliva anthemifolia* (Table 2). The top 10 species had significant differences in their proportions of the seed reserve, with *S. latifolia* accounting for 59.75%, *S. americanum* for 10.45%, and *P. clematidea* for 9.81% of the total seed reserves, and the top three species accounted for >80% of the total seed reserve, indicating that the soil seed bank of Dadu Taiwan is characterized by a small number of species occupying a large proportion of the seed reserve. The number of naturalized plant species was 40, accounting for 44.0% of the total species, but 90.9% of the total seed reserve. In the soil seed bank, there were three rare species listed in *The Red List of Vascular Plants of Taiwan, 2017*, which were the endangered species (EN) *Epaltes australis* (one seedling), the vulnerable species (VU) *Z. avicennae* (one seedling), and the data-deficient (DD) *A. pedunculata* (seven seedlings).

**Table 2.** Species composition of aboveground vegetation and soil seed banks among different vegetation types in Dadu Terrace of Taiwan (seeds/m$^2$).

| Species | Above Ground Vegetation [1] | Growth Type | Native/ Naturalized | Life Span [2] | *Panicum maximum* Type (*n* = 9) [3] | *Acacia confusa* Type (*n* = 20) | *Ficus microcarpa* Type (*n* = 6) | *Litsea glutinosa* Type (*n* = 6) | *Cinnamomum camphora* Type (*n* = 4) | *Trema orientalis* Type (*n* = 1) | Total | Ratio (%) |
|---|---|---|---|---|---|---|---|---|---|---|---|---|
| *Spermacoce latifolia* | + | Herb | Naturalized | A | 3739.9 | 4357.0 | 7437.0 | 9022.2 | 10940.7 | 1481.5 | 5756.5 | 59.75 |
| *Solanum americanum* | + | Herb | Naturalized | A | 2953.1 | 463.7 | 884.0 | 548.1 | 448.1 | 59.3 | 1006.4 | 10.45 |
| *Praxelis clematidea* | + | Herb | Naturalized | A | 2388.5 | 854.1 | 143.2 | 390.1 | 392.6 | 148.1 | 945.6 | 9.81 |
| *Panicum maximum* | + | Herb | Naturalized | P | 1121.0 | 739.3 | 145.7 | 118.5 | 322.2 | 148.1 | 606.4 | 6.29 |
| *Kyllinga brevifolia* | − | Herb | Native | P | 3.3 | 521.5 | 2.5 | 0.0 | 11.1 | 0.0 | 228.7 | 2.37 |
| *Trema orientalis* | + | Tree | Native | P | 6.6 | 93.3 | 64.2 | 74.1 | 66.7 | 5985.2 | 195.8 | 2.03 |
| *Conyza sumatrensis* | − | Herb | Naturalized | A | 261.7 | 15.6 | 49.4 | 434.6 | 0.0 | 0.0 | 121.1 | 1.26 |
| *Oplismenus compositus* | + | Herb | Native | P | 37.9 | 113.3 | 7.4 | 0.0 | 518.5 | 0.0 | 102.7 | 1.07 |
| *Mikania micrantha* | + | Vine | Naturalized | P | 14.8 | 68.9 | 22.2 | 113.6 | 118.5 | 666.7 | 75.4 | 0.78 |
| *Soliva anthemifolia* | − | Herb | Naturalized | A | 339.1 | 0.0 | 0.0 | 0.0 | 0.0 | 0.0 | 66.3 | 0.69 |
| *Acacia confusa* | + | Tree | Native | P | 0.0 | 123.7 | 17.3 | 61.7 | 14.8 | 0.0 | 65.4 | 0.68 |
| *Bidens pilosa* var. *radiata* | + | Herb | Naturalized | A | 174.5 | 42.2 | 19.8 | 12.3 | 40.7 | 118.5 | 62.8 | 0.65 |
| *Gnaphalium purpureum* | − | Herb | Naturalized | A | 107.0 | 29.6 | 29.6 | 39.5 | 11.1 | 0.0 | 43.8 | 0.45 |
| *Oxalis corniculata* | + | Herb | Native | A | 31.3 | 15.6 | 51.9 | 79.0 | 151.9 | 0.0 | 43.2 | 0.45 |
| *Centella asiatica* | + | Herb | Native | P | 0.0 | 1.5 | 192.6 | 4.9 | 118.5 | 0.0 | 36.7 | 0.38 |
| *Toddalia asiatica* | + | Vine | Native | P | 0.0 | 0.0 | 0.0 | 0.0 | 400.0 | 0.0 | 34.8 | 0.36 |
| *Acronychia pedunculata* | + | Tree | Native | P | 0.0 | 49.6 | 0.0 | 0.0 | 3.7 | 0.0 | 21.9 | 0.23 |
| *Urena lobata* | + | Shrub | Native | P | 1.6 | 40.7 | 9.9 | 2.5 | 7.4 | 0.0 | 20.3 | 0.21 |
| *Lindernia crustacea* | − | Herb | Native | A | 6.6 | 0.7 | 14.8 | 61.7 | 70.4 | 0.0 | 17.7 | 0.18 |
| *Youngia japonica* | + | Herb | Native | A | 46.1 | 10.4 | 0.0 | 0.0 | 0.0 | 0.0 | 42 | 0.14 |
| *Eleusine indica* | − | Herb | Native | A | 24.7 | 13.3 | 0.0 | 0.0 | 0.0 | 0.0 | 33 | 0.11 |
| *Polygonum chinense* | + | Herb | Native | A | 1.6 | 23.0 | 0.0 | 0.0 | 0.0 | 0.0 | 32 | 0.11 |
| *Cyperus compressus* | − | Herb | Native | A | 13.2 | 16.3 | 0.0 | 0.0 | 0.0 | 0.0 | 30 | 0.10 |
| *Cardamine flexuosa* | − | Herb | Naturalized | A | 0.0 | 18.5 | 4.9 | 4.9 | 0.0 | 0.0 | 29 | 0.10 |
| *Sida alnifolia* | − | Shrub | Native | P | 0.0 | 19.3 | 0.0 | 0.0 | 0.0 | 0.0 | 26 | 0.09 |
| *Pluchea sagittalis* | − | Herb | Naturalized | P | 13.2 | 5.9 | 0.0 | 19.8 | 7.4 | 0.0 | 26 | 0.09 |
| *Elephantopus mollis* | − | Herb | Naturalized | P | 1.6 | 14.1 | 0.0 | 0.0 | 0.0 | 29.6 | 22 | 0.07 |
| *Miscanthus floridulus* | + | Herb | Native | P | 0.0 | 2.2 | 4.9 | 2.5 | 48.1 | 0.0 | 19 | 0.06 |
| *Vernonia cinerea* | + | Herb | Native | A | 16.5 | 5.9 | 0.0 | 0.0 | 0.0 | 0.0 | 18 | 0.06 |
| *Rhynchelytrum repens* | + | Herb | Naturalized | P | 28.0 | 0.0 | 0.0 | 0.0 | 0.0 | 0.0 | 17 | 0.06 |
| *Phytolacca americana* | − | Herb | Naturalized | P | 23.0 | 1.5 | 0.0 | 0.0 | 0.0 | 0.0 | 16 | 0.05 |
| *Flueggea suffruticosa* | − | Shrub | Native | P | 0.0 | 9.6 | 0.0 | 0.0 | 0.0 | 0.0 | 13 | 0.04 |
| *Cyperus esculentus* | − | Herb | Naturalized | P | 4.9 | 0.0 | 7.4 | 17.3 | 0.0 | 0.0 | 13 | 0.04 |
| *Ixeris chinensis* | − | Herb | Native | P | 18.1 | 0.7 | 0.0 | 0.0 | 0.0 | 0.0 | 12 | 0.04 |
| *Chloris barbata* | − | Herb | Naturalized | P | 0.0 | 5.9 | 4.9 | 0.0 | 3.7 | 0.0 | 11 | 0.04 |

**Table 2.** *Cont.*

| Species | Above Ground Vegetation [1] | Growth Type | Native/ Naturalized | Life Span [2] | *Panicum maximum* Type (n = 9) [3] | *Acacia confusa* Type (n = 20) | *Ficus microcarpa* Type (n = 6) | *Litsea glutinosa* Type (n = 6) | *Cinnamomum camphora* Type (n = 4) | *Trema orientalis* Type (n = 1) | Total | Ratio (%) |
|---|---|---|---|---|---|---|---|---|---|---|---|---|
| *Boehmeria nivea* | − | Shrub | Naturalized | P | 0.0 | 5.2 | 0.0 | 0.0 | 14.8 | 0.0 | 11 | 0.04 |
| *Scoparia dulcis* | − | Herb | Naturalized | A | 0.0 | 0.0 | 0.0 | 9.9 | 25.9 | 0.0 | 11 | 0.04 |
| *Ipomoea obscura* | + | Vine | Naturalized | P | 0.0 | 3.0 | 0.0 | 0.0 | 0.0 | 88.9 | 10 | 0.03 |
| *Hedyotis corymbosa* | − | Herb | Native | A | 1.6 | 5.2 | 0.0 | 4.9 | 0.0 | 0.0 | 10 | 0.03 |
| *Conyza canadensis* | − | Herb | Naturalized | A | 13.2 | 0.0 | 0.0 | 2.5 | 0.0 | 0.0 | 9 | 0.03 |
| *Scleria terrestris* | − | Herb | Native | P | 0.0 | 4.4 | 7.4 | 0.0 | 0.0 | 0.0 | 9 | 0.03 |
| *Ficus microcarpa* | + | Tree | Native | P | 0.0 | 1.5 | 2.5 | 12.3 | 3.7 | 0.0 | 9 | 0.03 |
| *Ipomoea nil* | − | Vine | Naturalized | A | 0.0 | 5.9 | 0.0 | 0.0 | 0.0 | 0.0 | 8 | 0.03 |
| *Symplocos chinensis* | + | Shrub | Native | P | 0.0 | 0.7 | 2.5 | 14.8 | 0.0 | 0.0 | 8 | 0.03 |
| *Axonopus compressus* | − | Herb | Naturalized | P | 0.0 | 5.9 | 0.0 | 0.0 | 0.0 | 0.0 | 8 | 0.03 |
| *Zanthoxylum avicennae* | + | Tree | Native | P | 0.0 | 3.7 | 0.0 | 0.0 | 7.4 | 0.0 | 7 | 0.02 |
| *Mallotus japonicus* | + | Tree | Native | P | 0.0 | 5.2 | 0.0 | 0.0 | 0.0 | 0.0 | 7 | 0.02 |
| *Ficus subpisocarpa* | − | Tree | Native | P | 0.0 | 3.0 | 4.9 | 0.0 | 3.7 | 0.0 | 7 | 0.02 |
| *Ageratum houstonianum* | + | Herb | Naturalized | A | 0.0 | 3.0 | 4.9 | 0.0 | 0.0 | 0.0 | 6 | 0.02 |
| *Mimosa pudica* | + | Shrub | Naturalized | P | 0.0 | 3.7 | 0.0 | 0.0 | 0.0 | 0.0 | 5 | 0.02 |
| *Lantana camara* | + | Shrub | Naturalized | P | 0.0 | 2.2 | 0.0 | 4.9 | 0.0 | 0.0 | 5 | 0.02 |
| *Morus alba* | + | Shrub | Native | P | 1.6 | 3.0 | 0.0 | 0.0 | 0.0 | 0.0 | 5 | 0.02 |
| *Tephrosia noctiflora* | − | Herb | Naturalized | A | 6.6 | 0.0 | 0.0 | 0.0 | 0.0 | 0.0 | 4 | 0.01 |
| *Lepidagathis inaequalis* | − | Herb | Native | P | 1.6 | 1.5 | 2.5 | 0.0 | 0.0 | 0.0 | 4 | 0.01 |
| *Litsea glutinosa* | + | Shrub | Naturalized | P | 0.0 | 0.0 | 0.0 | 9.9 | 0.0 | 0.0 | 4 | 0.01 |
| *Mussaenda parviflora* | + | Vine | Native | P | 0.0 | 2.2 | 2.5 | 0.0 | 0.0 | 0.0 | 4 | 0.01 |
| *Hedyotis dichotoma* | − | Herb | Native | A | 0.0 | 0.7 | 0.0 | 7.4 | 0.0 | 0.0 | 4 | 0.01 |
| *Drymaria diandra* | − | Herb | Naturalized | A-P | 0.0 | 3.0 | 0.0 | 0.0 | 0.0 | 0.0 | 4 | 0.01 |
| *Ixeris polycephala* | − | Herb | Native | A | 4.9 | 0.0 | 0.0 | 0.0 | 0.0 | 0.0 | 3 | 0.01 |
| *Broussonetia papyrifera* | + | Tree | Native | P | 0.0 | 1.5 | 2.5 | 0.0 | 0.0 | 0.0 | 3 | 0.01 |
| *Chromolaena odorata* | + | Herb | Naturalized | P | 1.6 | 0.0 | 2.5 | 2.5 | 0.0 | 0.0 | 3 | 0.01 |
| *Rubus parvifolius* | − | Shrub | Native | P | 0.0 | 0.0 | 4.9 | 0.0 | 0.0 | 0.0 | 2 | 0.01 |
| *Liquidambar formosana* | − | Tree | Native | P | 1.6 | 0.0 | 0.0 | 2.5 | 0.0 | 0.0 | 2 | 0.01 |
| *Cyperus cyperoides* | − | Herb | Native | P | 3.3 | 0.0 | 0.0 | 0.0 | 0.0 | 0.0 | 2 | 0.01 |
| *Melochia corchorifolia* | − | Shrub | Naturalized | P | 0.0 | 0.7 | 2.5 | 0.0 | 0.0 | 0.0 | 2 | 0.01 |
| *Solanum trianthum* | − | Shrub | Naturalized | P | 1.6 | 0.7 | 0.0 | 0.0 | 0.0 | 0.0 | 2 | 0.01 |
| *Maesa perlaria* var. *formosana* | − | Shrub | Native | P | 0.0 | 1.5 | 0.0 | 0.0 | 0.0 | 0.0 | 2 | 0.01 |
| *Macaranga tanarius* | + | Tree | Native | P | 1.6 | 0.7 | 0.0 | 0.0 | 0.0 | 0.0 | 2 | 0.01 |

**Table 2.** *Cont.*

| Species | Above Ground Vegetation [1] | Growth Type | Native/ Naturalized | Life Span [2] | *Panicum maximum* Type (*n* = 9) [3] | *Acacia confusa* Type (*n* = 20) | *Ficus microcarpa* Type (*n* = 6) | *Litsea glutinosa* Type (*n* = 6) | *Cinnamomum camphora* Type (*n* = 4) | *Trema orientalis* Type (*n* = 1) | Total | Ratio (%) |
|---|---|---|---|---|---|---|---|---|---|---|---|---|
| *Alpinia zerumbet* | + | Herb | Native | P | 0.0 | 1.5 | 0.0 | 0.0 | 0.0 | 0.0 | 2 | 0.01 |
| *Dianella ensifolia* | + | Herb | Native | P | 0.0 | 1.5 | 0.0 | 0.0 | 0.0 | 0.0 | 2 | 0.01 |
| *Passiflora suberosa* | + | Vine | Naturalized | P | 0.0 | 0.7 | 0.0 | 2.5 | 0.0 | 0.0 | 2 | 0.01 |
| *Bridelia monoica* | + | Tree | Native | P | 0.0 | 0.7 | 0.0 | 0.0 | 0.0 | 0.0 | 1 | <0.01 |
| *Sarcandra glabra* | − | Shrub | Native | P | 0.0 | 0.7 | 0.0 | 0.0 | 0.0 | 0.0 | 1 | <0.01 |
| *Cinnamomum camphora* | + | Tree | Native | P | 0.0 | 0.0 | 0.0 | 0.0 | 3.7 | 0.0 | 1 | <0.01 |
| *Crotalaria zanzibarica* | + | Shrub | Naturalized | A | 0.0 | 0.7 | 0.0 | 0.0 | 0.0 | 0.0 | 1 | <0.01 |
| *Sapium sebiferum* | + | Tree | Naturalized | P | 0.0 | 0.7 | 0.0 | 0.0 | 0.0 | 0.0 | 1 | <0.01 |
| *Mallotus repandus* | + | Vine | Native | P | 0.0 | 0.7 | 0.0 | 0.0 | 0.0 | 0.0 | 1 | <0.01 |
| *Fimbristylis aestivalis* | − | Herb | Native | A | 1.6 | 0.0 | 0.0 | 0.0 | 0.0 | 0.0 | 1 | <0.01 |
| *Polygonum plebeium* | − | Herb | Naturalized | A | 0.0 | 0.0 | 0.0 | 2.5 | 0.0 | 0.0 | 1 | <0.01 |
| *Epaltes australis* | − | Herb | Native | A | 1.6 | 0.0 | 0.0 | 0.0 | 0.0 | 0.0 | 1 | <0.01 |
| *Morinda parvifolia* | + | Vine | Native | P | 0.0 | 0.0 | 0.0 | 2.5 | 0.0 | 0.0 | 1 | <0.01 |
| *Melia azedarach* | + | Tree | Native | P | 0.0 | 0.0 | 2.5 | 0.0 | 0.0 | 0.0 | 1 | <0.01 |
| *Gnaphalium purpureum* | − | Herb | Naturalized | A | 0.0 | 0.0 | 2.5 | 0.0 | 0.0 | 0.0 | 1 | <0.01 |
| *Lophatherum gracile* | + | Herb | Native | P | 0.0 | 0.7 | 0.0 | 0.0 | 0.0 | 0.0 | 1 | <0.01 |
| *Clerodendrum cyrtophyllum* | + | Shrub | Native | P | 0.0 | 0.7 | 0.0 | 0.0 | 0.0 | 0.0 | 1 | <0.01 |
| *Sonchus oleraceus* | − | Herb | Naturalized | A | 0.0 | 0.0 | 0.0 | 2.5 | 0.0 | 0.0 | 1 | <0.01 |
| *Momordica charantia* var. *abbreviata* | + | Vine | Naturalized | A | 0.0 | 0.0 | 0.0 | 2.5 | 0.0 | 0.0 | 1 | <0.01 |
| *Pericampylus glaucus* | + | Vine | Native | P | 0.0 | 0.7 | 0.0 | 0.0 | 0.0 | 0.0 | 1 | <0.01 |
| *Ageratum conyzoides* | − | Herb | Naturalized | A | 0.0 | 0.0 | 2.5 | 0.0 | 0.0 | 0.0 | 1 | <0.01 |
| *Bothriospermum zeylanicum* | − | Herb | Native | A | 0.0 | 0.0 | 0.0 | 2.5 | 0.0 | 0.0 | 1 | <0.01 |
| *Duchesnea indica* | − | Herb | Naturalized | P | 1.6 | 0.0 | 0.0 | 0.0 | 0.0 | 0.0 | 1 | <0.01 |
| Total | | | | | 11,422.2 | 7754.8 | 9160.5 | 11,093.8 | 13,755.6 | 8725.9 | 29,914 | 100.00 |

[1] Aboveground vegetation: +, present; −, absent. [2] Life span: A, annual–biennial; P, perennial. [3] *n*: represents the number of plots.

The dominant species in the seed reserve of the soil seed bank varied by vegetation type, where *O. compositus* was significantly more abundant in *Cinnamomum camphora* type; *P. clematidea*, *S. americanum*, and *P. maximum* were significantly more abundant in terms of quantity in the *Panicum maximum* type; the seed reserves of *T. orientalis* and *M. micrantha* were mainly found in *Trema orientalis* type; and *K. brevifolia* was more abundant in the *Acacia confusa* type. With the exception of the *Trema orientalis* type, the seed reserves of all vegetation types were dominated by herbs, and seed reserves were mainly dominated by a few species (Table 2). There was no significant difference in the number of plant species and seed reserves among the vegetation types in the soil seed bank (Table 1). The post hoc results of pairwise comparison of vegetation types showed that the number of tree species differed significantly ($p < 0.05$). The *Panicum maximum type* was significantly less than the other three vegetation types, including *Acacia confusa* type, *Ficus microcarpa* type, and *Litsea glutinosa* type. However, the tree seed reserves only differed significantly between *Panicum maximum* type and *Acacia confusa* type ($p < 0.05$). The differences among the other vegetation types were not significant (Table 1). Furthermore, our result showed that the tree species and seed reserves in the soil seed bank of *Panicum maximum* type at Dadu Terrace were the lowest.

The proportion of naturalized plant species in the soil seed bank was between 52 and 73% for each vegetation type, except for *Trema orientalis* type. Among them, the percentage of naturalized plant species in *Panicum maximum* type was significantly higher than that in *Acacia confusa* type ($p < 0.05$) (Table 1). The proportion of naturalized plants in the seed reserve of *Trema orientalis* type (31.4%) was lower, whereas the naturalized plant seed reserves of the other vegetation types ranged from 78.0 to 97.3%. The naturalized plant proportions of the seed reserve were not significantly different among vegetation types (Table 1).

The DCA results of the 46 plots of soil seed banks in Dadu Terrace (Figure 7) showed that the total variation was 3.641; the characteristic values of the first three axes were 0.668, 0.376, and 0.219, where the explanation values of variation were 18.34, 10.33, and 6.02% and the lengths of the axes were 3.38, 2.47, and 2.28, respectively. In the ordination diagram, the sampling plots of *Cinnamomum camphora* type and *Litsea glutinosa* type were distributed on the left side of axis 1, while the sampling plots of *Acacia confusa* type, *Ficus microcarpa* type, and *Panicum maximum* type were grouped separately on axis 1, showing that *Cinnamomum camphora* type and *Litsea glutinosa* type have a similar composition of soil seed banks. *Acacia confusa* type, *Ficus microcarpa* type, and *Panicum maximum* type were affected by the difference in the geographical location of the sampling plot, and the composition of the soil seed bank was more variable. Axis 2 separated *Trema orientalis* type and *Panicum maximum* type, showing the difference in the composition of the two vegetation types.

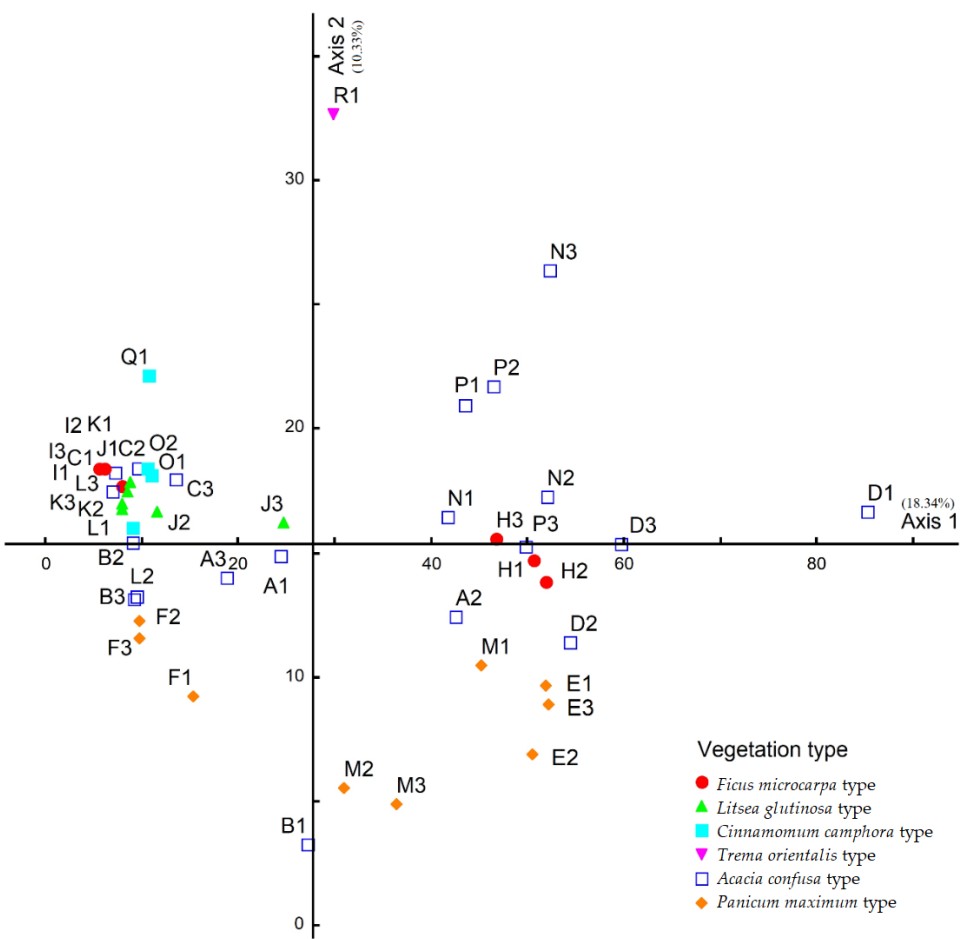

**Figure 7.** Detrended correspondence analysis (DCA) of the soil seed bank in Dadu Terrace, Taiwan.

*3.3. Similarities between Soil Seed Bank and Aboveground Plant Composition*

A total of 57 families, 147 genera, and 185 species of spermatophytes were investigated in this study, of which 47 species occurred both in the aboveground vegetation and in the soil seed bank. The overall Sørensen similarity index between the aboveground vegetation type and soil seed bank was 50.1%, with *Litsea glutinosa* type being the highest (0.50 ± 0.12) and *Ficus microcarpa* type being the lowest (0.16 ± 0.05). However, the variation in Sørensen similarity indices varied widely among sampling plots of different aboveground vegetation types (Table 1).

The analysis of the first two axes using PCoA revealed that the species compositions of the aboveground vegetation type and the soil seed bank were separated by axis 1, indicating there was a greater variation in the species composition between each other (Figure 8). The PCoA result of the compositions between aboveground vegetation and the soil seed bank showed that the explanation rates of variation in the first three axes were 28.93, 9.92, and 7.63%, respectively. Axis 2 mostly showed similarities in the composition of aboveground and soil seed banks, i.e., the sampling plots of *Acacia confusa* type containing *A. pedunculata* (N1, N2, N3, P1, P2, and P3) were mainly distributed above axis 2, whereas the sampling plots of *Panicum maximum* type (E1, E2, E4, M1, M2, and M3) were mainly found below axis 2, which showed the correlation between the composition of aboveground vegetation species and the soil seed bank.

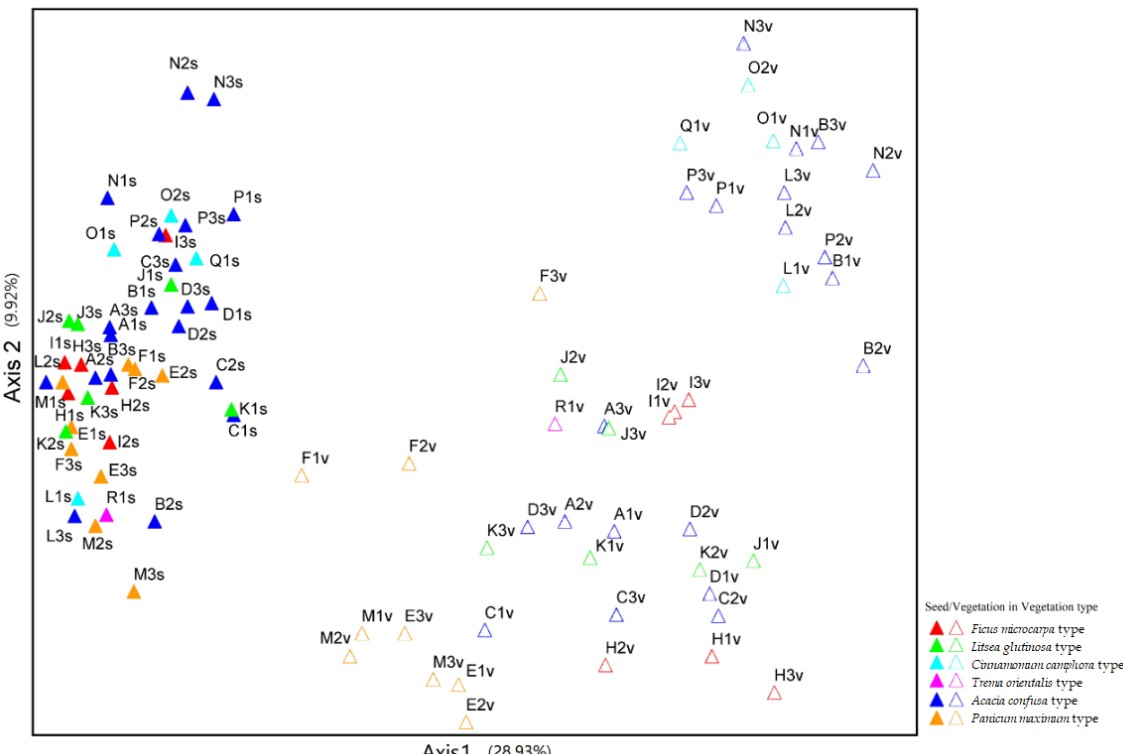

**Figure 8.** Principal coordinate analysis (PCoA) plot of species composition between aboveground vegetation and soil seed bank in Dadu Terrace, Taiwan.

## 4. Discussion

### 4.1. Composition Characteristics of Aboveground Vegetation and Soil Seed Bank

The Dadu Terrace forest is an essential urban forest in central Taiwan and is crucial for landscaping, environmental purification, and public recreation. Due to cemeteries surrounding the forests, people pay respect to the dead at these tombs according to traditional customs and clean the weeds surrounding the cemetery, which often leads to wildfire accidents. There were 3418 fires in Dadu Terrace from 2011 to 2013 [32], with the highest percentage of fires caused by *P. maximum* weeds. The period of greatest fire occurrence was from late autumn of every year to April of the following year (the Qingming tomb-sweeping period of the Han people). Under these highly frequent fire disturbances, the grassland composed of native *M. sinensis* has been gradually replaced by *P. maximum* and other exotic naturalized plants [9]. The aboveground composition of the grassland comprises exotic plants adapted to highly frequent fires such as *P. maximum*, *L. camara*, and other postfire-emergent plants or annual and biennial plants such as *B. pilosa* var. *radiata* and *P. clematidea*, which have further formed a dominant grassland with plants such as *P. maximum*. This reflects the fact that Dadu Terrace is heavily invaded by naturalized plants such as *P. maximum*, making the number of invasive species and percentage of ground coverage higher than those in the forest area, along with crowding out the survival space of native species and reducing species diversity. The CCA ordination diagram shows that nutrients in grasslands, including soil nitrogen and cations, decreased, and soil pH increased, altering the pH and other properties of soil that were now not conducive to the succession of the forest.

The proportion of naturalized plants soil seed banks at Dadu Terrace ranged from 52.4 to 90.0%, and that of the seed reserves ranged from 31.4 to 97.3%, values which are much higher than those in other areas of Taiwan [20–22,52]. This phenomenon indicates that naturalized plants have become the dominant seed reserve component of the sustainable soil seed bank at Dadu Terrace. Aboveground cover compositions with higher proportions of naturalized plants also have higher proportions in the soil seed bank, reflecting the

influence of naturalized plants in aboveground vegetation on soil seed bank composition [53–55]. The causes of this phenomenon are mainly historical factors of the region, such as past fire disturbance, afforestation, and agricultural activities.

Frequent fire disturbances can cause different effects. Some cases show seed germination out of the top layer of soil seed bank, reducing the species diversity and seed reserve [56,57] but some do not [58]. In recent years, the frequent fires on the grasslands of Dadu Terrace have led to the deterioration of the species composition and structure of the aboveground vegetation. Only the plants adapted to the high frequency of fires have established their populations. Frequent fires result in a higher proportion of seeds in the soil seed bank of plants with shorter life histories or reproductive periods [59]. Plants adapted to frequent fires are mostly naturalized plants, resulting in a higher similarity between the aboveground vegetation and the seed plant composition of the soil seed bank [3,20,60]. Fire disturbance allows naturalized plants to maintain their dominance [61,62], and their species invest more resources in the root. It also alters local environmental systems' carbon and nitrogen cycles [63,64]. Some naturalized plants even increase the occurrence and frequency of fires in the area [65,66]. The interaction of fire disturbance and invasion by naturalized plants has resulted in the formation of *P. maximum* grassland at Dadu Terrace, which is dominated by invasive naturalized plants, resulting in a seed bank continuously characterized by the absolute dominance of exotic plants as seed reserves in the soil seed bank.

### 4.2. Feasibility of Soil Seed Banks for Vegetation Restoration

This study found that herbaceous plants dominated the seed reserve in the soil seed bank of the Dadu Terrace urban forest, where, except for *Trema orientalis* type, naturalized plants accounted for >70% of the seed reserves in the soil seed bank of each vegetation type. Naturalized plants have formed a continuous soil seed bank at Dadu Terrace and are seriously affecting the composition and reserves of the seed bank. In addition, the woody plant species and their seed reserve of *Panicum maximum* type grasslands were lower than those of other forest vegetation types, indicating that the composition of the soil seed bank of *P. maximum* grasslands is highly unfavorable to the natural restoration of urban forests. The main reason for this phenomenon is the excessive fire disturbance, which prevents the survival of postemergence seedlings and young trees, whereas annual plants such as *S. latifolia*, *S. americanum*, and *P. clematidea*, or perennials such as *P. maximum* and *L. camara* have better adaptability to frequent fires, thus forming the current plant community that is dominated by *P. maximum*. Although urban forest ecological restoration is conducted in Dadu Terrace through plantation construction, for planted forests such as *Acacia confusa* type, *Cinnamomum camphora* type, *Litsea glutinosa* type, and *Ficus microcarpa* type, and secondary forest *Trema orientalis* type, the number of species and seed reserves of naturalized plants still accounts for a very high proportion of the soil seed bank.

The limitation of seed germination is also one of the limitations of forest colonization [67] and the seeds of trees such as *A. confusa* and *T. orientalis*, native pioneer species, break dormancy and leave the soil seed bank to grow into seedlings when stimulated and disturbed by light [68]. Preliminary tests showed that the germination of *A. confusa* seeds were higher after soil disturbance than without disturbance by fire in *P. maximum* grasslands at Dadu Terrace [3], suggesting that artificial soil disturbance could increase the germination of *A. confusa* seedlings in the soil seed bank of *P. maximum* grasslands. However, *A. confusa* seedlings grow more slowly than herbaceous plants and therefore require proper management such as mowing and vine removal to grow into young trees taller than *P. maximum* [69], which is difficult for *A. confusa* to restore the forest on its own.

The area of *Trema orientalis* type was relatively small at Dadu Terrace and was a secondary forest at the beginning of forest succession; its soil seed bank was the only sampling plot in this study that was dominated by tree growth types. The drupes of *T. orientalis* are bird feeding [70], and the seeds are suborthodox [71] and are commonly found in the soil seed banks in low-elevation forests in Taiwan [3,16,20–22,30,52]. *T. orientalis* is

also a fast-growing pioneer species common after forest disturbances in tropical forests in Asia [2,72–75], and is a species with forest restoration potential [76]. In the *Trema orientalis* type at Dadu Terrace, due to the presence of *T. orientalis* parent trees and the fact that *T. orientalis* seeds need to be stimulated by variable temperature or light to break dormancy and germinate out of the soil [68], *T. orientalis* seeds can continuously accumulate in the soil seed bank under the shade of their parent tree.

In order to restore and improve its forest ecosystem services in the Dadu Terrace urban forest, it would be ineffective to wait for natural restoration or negative strategies such as making appeals or signs to advise people to reduce disruption. Significantly, invasive plants have severely altered the native vegetation form and it is difficult for forest seeds to colonize in the adjacent urban forest. Moreover, because almost the entire Dadu Terrace forest is planted, the species composition is relatively simple and naturalized plants are abundant. Furthermore, although there are 14 native tree species in the soil seed bank, the main tree seed reserve is mainly dominated by *T. orientalis* of *Trema orientalis* type. Compared with the neighboring Dakeng area, the native species and species reserve of the Dadu Terrace are significantly insufficient [30]. Therefore, the Dakeng area would be used as a template for developing the Dadu Terrace ecological restoration project by referring to the composition of reference ecosystems in similar environments of the neighboring areas. This template was used as a benchmark to evaluate the restoration project's effectiveness in the later stages [75,77].

In addition to accelerating forest restoration through afforestation or propagating the introduction of native species [78–83], the use of soil seed banks to achieve vegetation restoration is also an ecological restoration strategy [84–86], i.e., to increase species diversity by transferring forest-seed-rich soils to sites where forest seeds are scarce or where soils are degraded. In this study, only *Trema orientalis* type was a secondary forest in Dadu Terrace (however, the area was minimal), and its soil seed bank was the only plant community that was dominated by native tree seed reserves. However, it had the potential for soil transfer, only one single species—*T. orientalis*—in terms of the richness of tree species was in its soil seed bank, which needs to be considered. Therefore, if Dadu Terrace were to use a soil seed bank for soil transfer restoration, it is recommended to consider soils from neighboring areas such as the Dakeng area in Taichung City, where there are more native species [2,30], for its advantages of rich species diversity and a high proportion of native species. By integrating ecological reforestation and soil seed bank restoration, we can accelerate establishing a forest ecosystem close to nature and compatible with urban forest functions. However, in addition to human intervention to accelerate the effectiveness of restoration, the reduction in fire disturbance is the key to the success of ecological restoration in Dadu Terrace.

## 5. Conclusions

The Dadu Terrace soil seed bank is mainly composed of herbaceous plants, of which naturalized plants account for >90% of the seed reserves. The tree seed reserves in the soil seed bank are mostly found in *Trema orientalis* type and *Acacia confusa* type, where *T. orientalis* and *A. confusa* are the most abundant tree species in the soil seed bank. However, due to frequent fire disturbance and light stimulation, seeds of *T. orientalis* and *A. confusa* tend to germinate and reduce the seed reserves of the soil seed bank, and their seedlings and young trees have difficulty surviving in the *Panicum maximum* type. In the future, it will be challenging for the *Panicum maximum* type to use the soil seed bank for successful restoration of the forest after the disturbance, and the restoration of the forest depends on moderate human intervention. The amount of *T. orientalis* in the soil seed banks was more than 20 times higher than that of parent tree vegetation types, and it was the only one in Dadu Terrace where trees and native species dominate the proportion of seed reserves, and thus has a high value for soil transfer. However, just one tree species—*T. orientalis*—is an insufficient diversity of tree species. If the soil seed bank transfer methodology is suited for

restoration of species diversity and reforestation, it is recommended to use the soils from neighboring areas with more native and local species as Dakeng area.

**Author Contributions:** Conceptualization, C.-Y.L. and H.-Y.T.; methodology, C.-Y.L., W.W. and H.-Y.T.; software, C.-Y.L., M.-C.L. and W.W.; validation, C.-Y.L., M.-C.L. and H.-Y.T.; formal analysis, C.-Y.L. and W.W.; investigation, C.-Y.L. and W.W.; resources, C.-Y.L. and H.-Y.T.; data curation, C.-Y.L., M.-C.L., W.W. and H.-Y.T.; writing—original draft preparation, C.-Y.L. and H.-Y.T.; writing—review and editing, C.-Y.L., M.-C.L., W.W. and H.-Y.T.; visualization, C.-Y.L., M.-C.L., W.W. and H.-Y.T.; supervision, H.-Y.T.; project administration, H.-Y.T.; funding acquisition, H.-Y.T. All authors have read and agreed to the published version of the manuscript.

**Funding:** This research received no external funding.

**Institutional Review Board Statement:** This research did not involve humans or animals.

**Informed Consent Statement:** This research did not involve humans or animals.

**Data Availability Statement:** Data are available upon request.

**Acknowledgments:** We are grateful to the Author Services of MDPI for the English language editing. We would also like to aknowledge the Lab of Plant Taxonomy and Forest Ecology, Department of Forestry, National Chung-Hsing University (NCHU), in the field investigation.

**Conflicts of Interest:** The authors declare no conflict of interest.

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
