# Peer review of "Composition Characteristics of an Urban Forest Soil Seed Bank and Its Influence on Vegetation Restoration: A Case Study in Dadu Terrace, Central Taiwan"

_sustainability, doi:10.3390/su14074178_

Round 1

Reviewer 1 Report

Introduction: In could be more focussed with the available relevant literatures.

Materials and methods: All the statis analysis should be made more clear alsong with software used for analysis

Details about replicates should be provided

Result: Written well abut details about findings are missing in some of the sections. They should be provided

Discussion: Need to be improved to make it more focussed with emphasis on the novelty of the findings 

Author Response

Reply to Reviewer 1:

Thank you very much for your valuable comments.

Introduction: It could be more focussed with the available relevant literatures.

  • We have added more existing literature.

Materials and methods: All the statis analysis should be made more clear along with software used for analysis.

  • Statistics and analysis software are also described.

Details about replicates should be provided.

  • The discussion has also been partially revised.

Result: Written well abut details about findings are missing in some of the sections. They should be provided.

Discussion: Need to be improved to make it more focussed with emphasis on the novelty of the findings.

  • Could you be specific about the missing parts of the results in which part?

Reviewer 2 Report

Urban forests and green space protection are very important for biodiversity protection, human settlements improvement and vegetation restoration, and related research has attracted more and more attention. Based on the data collected from 46 sample plots in Dadu Terrace, Central Taiwan, this paper studies the composition of urban forest soil seed bank and its impact on vegetation restoration, and finally puts forward relevant suggestions for vegetation restoration. The paper has certain significance for scientific research and practical application. I personally gave some suggestions for the author to improve the article.

  • Contents of the introduction mainly introduces the relevant situation of the study area, rather than the relevant research progress. It is suggested to move some of the current content to the Study Site, and add relevant research progress in the introduction, including the research results in other areas and existing disputes around the theme of the article, so as to further reflect the scientific value of this study.
  • There are few references in the past five years. It is suggested that the author further collect the relevant research literature in recent years, and then analyze the scientific problems.
  • From the analysis of results, the impact of fire on soil seed bank and vegetation restoration in the study area is very important. Therefore, the proposed artificial restoration is only applicable to this area. It is suggested that the author further elaborate and explain this. In addition, the relationship between soil seed bank and aboveground vegetation restoration is affected by many factors. Therefore, it is suggested to further explain the conclusion of the article.
  • The title of figure 8 should be checked.

Author Response

Reply to Reviewer 2:

Thank you for your valuable comments.

Urban forests and green space protection are very important for biodiversity protection, human settlements improvement and vegetation restoration, and related research has attracted more and more attention. Based on the data collected from 46 sample plots in Dadu Terrace, Central Taiwan, this paper studies the composition of urban forest soil seed bank and its impact on vegetation restoration, and finally puts forward relevant suggestions for vegetation restoration. The paper has certain significance for scientific research and practical application. I personally gave some suggestions for the author to improve the article.

Contents of the introduction mainly introduces the relevant situation of the study area, rather than the relevant research progress. It is suggested to move some of the current content to the Study Site, and add relevant research progress in the introduction, including the research results in other areas and existing disputes around the theme of the article, so as to further reflect the scientific value of this study.

  • This article is mainly used to announce the current situation of the soil seed bank in the Dadu Terrace and to encourage the need for active human intervention in ecological restoration.

There are few references in the past five years. It is suggested that the author further collect the relevant research literature in recent years, and then analyze the scientific problems.  

  • We have added several recent literatures.

From the analysis of results, the impact of fire on soil seed bank and vegetation restoration in the study area is very important. Therefore, the proposed artificial restoration is only applicable to this area. It is suggested that the author further elaborate and explain this. In addition, the relationship between soil seed bank and aboveground vegetation restoration is affected by many factors. Therefore, it is suggested to further explain the conclusion of the article.

  • We had revised several paragraphs from "Introduction" to "study site". Other errors and inadequacies have also been revised.

The title of figure 8 should be checked. 

  • The title of figure 8 was revised.

Reviewer 3 Report

The authors evaluated the composition of the aboveground plant communities and the belowground soil seed bank in the grassland and forests in the urban areas. The findings that low species number and seed abundance of tree species in the soil seed bank can provide references for urban forest succession and management practices. Specific comments are listed below:

  1. In the Abstract, delete “We surveyed vegetation and soil seed banks and divided them into forests and grasslands by vegetation appearance.”; “via the germination method”.
  2. The authors stated that “Even if the frequency of fires reduced, the forest would take a long time to recover itself.” The statement is not straightly derived from the results of current study, and more like the contents in the Discussion.
  3. In the Introduction, sentences such as “Restoration is the process of guiding and aiding the restoration of abiotic and biotic components of the environment to their undamaged or original state. The first step in any restoration plan is to understand the extent, severity, and restorability of the degradation of the site. “; ” The soil seed bank is the collection of seeds that have fallen to the surface of the soil through various dispersal mechanisms, excluding those that have been eaten by animals, lost their activity, decayed, or germinated, and the remaining seeds that are still active in the soil are grouped into a collection.” are general common sense and can be deleted.
  4. The information of sampling plots should be detailed. Such as, how many plots were set up? The distance between the sampling plots.
  5. Diameter at breast height (DBH) could not be applied on herbaceous plants.
  6. In section 2.3, the authors should define “quadrants”.
  7. Delete the 1st sentence of section 3.1.
  8. Change “The species diversity was compared among the aboveground vegetation types using the nonparametric Kruskal–Wallis test, and it was found that the number of species differed significantly (p < 0.05).” to “the number of species differed significantly among the aboveground vegetation types (p < 0.05)”.
  9. In Table 1, change “seeds/m-2” to “seeds/m2”.
  10. The sentence “and the number of tree species and tree seed reserve among the vegetation types were significantly different by growth type” is confused to me.
  11. The title of Figure 8 should be revised.
  12. Section 4.1, change “crucialfor” to “crucial for”.
  13. How the results of the CCA ordination diagram (Figure 6) can reflect the grasslands of Dadu Terrace had been affected by frequent fires?
  14. Change “”indicats” to “indicates”.
  15. The sentence “Fire disturbance helps naturalized plants ·····and increases the occurrence and frequency of fires” is difficult to understand to me.
  16. What does “negative strategies” refer to?

Author Response

Reply to Reviewer 3:

Thank you for the explicit comments listed. Most comments had revised in the manuscript with "track changes." There are a few points we want to reply.

The authors evaluated the composition of the aboveground plant communities and the belowground soil seed bank in the grassland and forests in the urban areas. The findings that low species number and seed abundance of tree species in the soil seed bank can provide references for urban forest succession and management practices. Specific comments are listed below:

  1. In the Abstract, delete “We surveyed vegetation and soil seed banks and divided them into forests and grasslands by vegetation appearance.”; “via the germination method”.
  • Down.
  1. The authors stated that “Even if the frequency of fires reduced, the forest would take a long time to recover itself.” The statement is not straightly derived from the results of current study, and more like the contents in the Discussion.
  • We delete the sentence.
  1. In the Introduction, sentences such as “Restoration is the process of guiding and aiding the restoration of abiotic and biotic components of the environment to their undamaged or original state. The first step in any restoration plan is to understand the extent, severity, and restorability of the degradation of the site. “; ” The soil seed bank is the collection of seeds that have fallen to the surface of the soil through various dispersal mechanisms, excluding those that have been eaten by animals, lost their activity, decayed, or germinated, and the remaining seeds that are still active in the soil are grouped into a collection.” are general common sense and can be deleted.
  • Down.
  1. The information of sampling plots should be detailed. Such as, how many plots were set up? The distance between the sampling plots.
  • Thanks for your suggestion. We added the detailed information for our sampling plot: "Study site was divided into three parts: north, central and south rigion by geography of Dadu Terrace. Each region selected several plant communities by vegetation composition and physiognomy characteristics, and 17 areas (A-R) were set up. Three plots were set up for each area, besides O_(1 plot), Q_(1 plot) and R_area (two plots) is due to the small size of the plant community. We set 46 sampling plots, and the distance between plots was about 10-30 m within the area.
  1. Diameter at breast height (DBH) could not be applied on herbaceous plants.
  • We modified the sentence as The sampling plot includes the frequency, coverage area, and basal area of plant occurrence. We recorded herbaceous plants, vines, and woody plants (with a diameter at breast height (DBH) <1 cm) as an understory to count their coverage area. The forest sampling plot was also surveyed for overstory of trees with DBH of >1 cm; its DBH was recorded, and basal area was calculated.
  1. In section 2.3, the authors should define “quadrants”.
  • Suppose the word “quadrant” is easily misunderstood. Quadrant is a word we use when drawing the horizontal axis (x-axis) and the vertical axis (y-axis). Two axes divided into four parts. Each part is called a quadrant. We have added additional information without affecting the semantics and content.
  1. Delete the 1st sentence of section 3.1.
  • Down.
  1. Change “The species diversity was compared among the aboveground vegetation types using the nonparametric Kruskal–Wallis test, and it was found that the number of species differed significantly (p < 0.05).” to “the number of species differed significantly among the aboveground vegetation types (p < 0.05)”.
  • Down.
  1. In Table 1, change “seeds/m-2” to “seeds/m2”.
  • Down.
  1. The sentence “and the number of tree species and tree seed reserve among the vegetation types were significantly different by growth type” is confused to me.
  • We want to emphasize that there is no difference when considering all plants, but only look at tree plants; there is a difference. We have revised the manuscript hope it will not be misunderstood.
  1. The title of Figure 8 should be revised.
  • Down.
  1. Section 4.1, change “crucialfor” to “crucial for”.
  • Down.
  1. How the results of the CCA ordination diagram (Figure 6) can reflect the grasslands of Dadu Terrace had been affected by frequent fires?
  • The frequent fire disturbance has caused a reduction in the forest area of Dadu Terrace. The CCA ordination diagram (Figure 6) show that nutrients in grasslands, including soil nitrogen and cations decreased, and soil pH increased. Additionally, maximum overgrows after the fire. So we were revised "The CCA ordination diagram (Figure 6) shows that nutrients in grasslands, including soil nitrogen and cations, decreased, and soil pH increased, altering the pH and other properties of soil that were now not conducive to the succession of the forest." in the discussion part of the manuscript.
  1. Change “”indicats” to “indicates”.
  • Down.
  1. The sentence “Fire disturbance helps naturalized plants ·····and increases the occurrence and frequency of fires” is difficult to understand to me.
  • The sentence was modified as “Fire disturbance keeps naturalized plants to maintain their dominance, which species invest more resources in the root. It also alters local environmental systems' carbon and nitrogen cycles.”
  1. What does “negative strategies” refer to?
  • When we mention “negative strategies,” People might make appeals or signs to advise people to reduce the disruption that the forest naturally recovers itself without human intervention.

Round 2

Reviewer 1 Report

The suggested revisions included and can be accepted for publication

Author Response

Reply to Reviewer :

General

-Next time do send also a version without track changes to facilitate reviewers.

-English revision is necessary.

  • Thank you for your comment. We have summitted a version without track changes to the Journal system.

Introduction

  1. i) Please do introduce the topic in general terms and then report specific information for Taiwan. I think that the first part of the introduction should focus on the importance of seed banks for plant community restoration in urban areas. I suggest to move the first paragraph (that focus on Taiwan) before describing objectives.
  • Thank you for your comment. We revised the order of two paragraphs and corrected the order of citations in the Introduction Section.
  1. ii) The first sentence related to objectives has a repetition and is incorrect. "In this study, we want to understand the composition characteristics of urban forest soil seed banks and their influence on vegetation restoration investigated the composition of the aboveground vegetation and soil seed bank of the Dadu Terrace"
  • Down.

Methods

  1. i) “10x10m”, “5X5m”
  • Down.
  1. ii) “covered area” rather than “coverage area”
  • Down.

iii) In general the following sentence should be revised “The sampling plot includes the frequency, coverage area, and basal area of plant occurrence.We recorded herbaceous plants, vines, and woody plants (with a diameter at breast height (DBH) <1 cm) as an understory to count their coverage area. The forest sampling plot was also surveyed for overstory of trees with DBH of >1 cm; its DBH was recorded, and basal area was calculated.”

  • Down.
  1. iv) Authors measured elevation and not altitude. Please, substitute “altitude” with “elevation” were necessary.
  • Down.
  1. v) Authors should report which are the physicochemical properties of soil that were measured. However, I suggest to remove this as such data is not part of the results.
  • Down.

Discussion

In this section authors should not indicate figures and tables (this reference should be in the results section).

  • Down.
